# Application-Aware Network Traffic Management in MEC-Integrated Industrial Environments

Paolo Bellavista [1,†] , Mattia Fogli [2,*,†,‡] , Carlo Giannelli [3,†,‡] and Cesare Stefanelli [2,†,‡]

1. Department of Computer Science and Engineering, University of Bologna, 40100 Bologna, Italy
2. Department of Engineering, University of Ferrara, 44122 Ferrara, Italy
3. Department of Mathematics and Computer Science, University of Ferrara, 44121 Ferrara, Italy
* Correspondence: mattia.fogli@unife.it
† These authors contributed equally to this work.
‡ This paper is an extended version of our paper published in WoWMoM 2022 as short paper: Fogli, M.; Giannelli, C.; Stefanelli, C. Joint Orchestration of Content-Based Message Management and Traffic Flow Steering in Industrial Backbones. In Proceedings of the 2022 IEEE 23rd International Symposium on a World of Wireless, Mobile and Multimedia Networks (WoWMoM), Belfast, UK, 14–17 June 2022; pp. 325–330, doi:10.1109/WoWMoM54355.2022.00067.

**Abstract:** The industrial Internet of things (IIoT) has radically modified industrial environments, not only enabling novel industrial applications but also significantly increasing the amount of generated network traffic. Nowadays, a major concern is to support network-intensive industrial applications while ensuring the prompt and reliable delivery of mission-critical traffic flows concurrently traversing the industrial network. To this end, we propose application-aware network traffic management. The goal is to satisfy the requirements of industrial applications through a form of traffic management, the decision making of which is also based on what is carried within packet payloads (application data) in an efficient and flexible way. Our proposed solution targets multi-access edge computing (MEC)-integrated industrial environments, where on-premises and off-premises edge computing resources are used in a coordinated way, as it is expected to be in future Internet scenarios. The technical pillars of our solution are edge-powered in-network processing (eINP) and software-defined networking (SDN). The concept of eINP differs from INP because the latter is directly performed on network devices (NDs), whereas the former is performed on edge nodes connected via high-speed links to NDs. The rationale of eINP is to provide the network with additional capabilities for packet payload inspection and processing through edge computing, either on-premises or in the MEC-enabled cellular network. The reported in-the-field experimental results show the proposal feasibility and its primary tradeoffs in terms of performance and confidentiality.

**Keywords:** application-aware network traffic management; edge computing; industrial Internet of things; in-network processing; multiaccess edge computing; software-defined networking

## 1. Introduction

In the past, industrial environments had almost no practical issues concerning the amount of network traffic produced or consumed at the shop floor level. Such traffic was fairly limited to mission-critical information generated by fixed equipment, primarily carrying operational and safety-related machine parameters. The network resources available onsite were (more than) adequate for managing it in a timely and reliable manner.

The advent of the Industrial Internet of Things (IIoT) is radically changing what modern industrial environments look like [1]. On the one hand, IIoT devices have enabled sophisticated applications (e.g., mobile asset tracking, online remote reconfiguration, and predictive maintenance) while being low in cost. On the other hand, they have made the shop floor more articulated than ever, not only challenging the network infrastructure with an unprecedented amount of traffic but also posing security threats [2]. In fact, IIoT devices

rely on complex chains of software dependencies (e.g., third-party libraries), which make integrity mechanisms difficult to be guaranteed from both an operational and a security perspective [3].

It is worth noting that the deluge of messages sent out by IIoT devices may clog up the network infrastructure. In addition, modern industrial equipment, e.g., Automated Guided Vehicles (AGVs) moving spare parts among the plant and the warehouse [4], may abruptly migrate traffic flows across different Network Devices (NDs), e.g., switches, routers, and access points, in relation to their physical position. As a result, it is paramount to manage non-mission-critical and mission-critical traffic dynamically. Otherwise, the non-mission-critical traffic could take the network resources at the expense of the mission-critical one. This leads to the following capabilities.

First, traffic flows should be properly prioritized: mission-critical flows go first, then non-mission-critical ones (flow prioritization). Secondly, traffic flows should be steered across the network according to their priority: mission-critical flows preempt the fastest routes whenever necessary (flow steering). Thirdly, information should be aggregated whenever possible while not infringing application-specific constraints, e.g., on update latency (data aggregation). Fourth, information should be discarded when carrying useless or not-so-relevant information (data filtering). Fifth, confidentiality should also be ensured for the traffic flows steered outside the trust boundaries of the industrial domain (data encryption).

Flow prioritization and steering are networking capabilities typically enforced by NDs against flows. Instead, data aggregation, filtering, and encryption are typically associated with more powerful computing capabilities. Although some rudimentary data aggregation and filtering forms could also be provided directly within NDs, such capabilities are usually provided at a decent level of expressiveness by general-purpose servers. The raison d'être of data aggregation and filtering is to reduce the traffic passing through the network. Thus, they should be performed as close as possible to data sources (i.e., the industrial equipment in our targeted scenarios) and whenever possible. However, industrial equipment typically has limited computing resources and, in most cases, cannot even host custom software due to warranty issues. This is where edge computing primarily and relevantly comes into play. Specifically, on-premises edge computing can provide such computing capabilities within the trust boundaries of the industrial domain. Because on-premises resources might not fit the dynamic demand in some cases (e.g., network traffic is temporarily too high for the employed industrial backbone), a practical countermeasure is to steer targeted traffic flows outside the industrial boundaries over cellular links, such as 5G. The integration of 5G with Multi-access Edge Computing (MEC) offers off-premises and elastically provisioned edge computing in the proximity of the industrial boundaries. The latency toward MEC nodes is significantly lower than it is toward the cloud, thus making the MEC a viable alternative even for demanding industrial scenarios. Accordingly, we jointly consider two flavors of edge computing for performing data aggregation, filtering, and encryption, i.e., on-premises edge computing and MEC. The former consists of Edge Nodes (ENs) deployed within the industrial boundaries and locally owned. In contrast, the latter consists of ENs deployed outside the industrial boundaries, on MEC resources owned by cellular telco operators in the industrial plant vicinity, made available in an on-demand cloud-like manner.

The data-encryption capability becomes fundamental as traffic may be steered outside the industrial boundaries over cellular links. Note that some legacy industrial machines could not enforce any security mechanism by design on their flows. Therefore, if such flows were redirected outside the industrial boundaries (e.g., to make room for mission-critical flows), they would be sent out as cleartext, breaking confidentiality. In this regard, the data-encryption capability can use on-premises ENs for ciphering data within the trust boundaries of the industrial domain (as close as possible to data sources) and subsequently transmit encrypted data only to the MEC nodes. MEC nodes can then decrypt such data through a/symmetric cryptography for further processing, e.g., data aggregation and

filtering. Another option is homomorphic cryptography [5], which allows MEC nodes to make simple operations on data with no need of first decrypting them.

Because payload inspection and processing (e.g., data aggregation, filtering, and encryption) may introduce additional latency, they should be enforced as much as possible without burdening mission-critical flows. To this purpose, flow prioritization should distinguish between mission-critical and non-mission-critical flows. Then, the flow steering capability is to (re)route flows down those paths fitting their application requirements and, along the way, possibly toward on-premises or off-premises ENs for further processing.

The overarching ambition is to satisfy the requirements of industrial applications through a form of traffic management, the decision-making of which is also based on what is carried within packet payloads (i.e., application data) rather than only on packet headers (mere forwarding). Accordingly, we propose a novel solution that provides the five capabilities outlined above as the basis for application-aware network traffic management. The proposed solution relies on two pillars: edge-powered In-Network Processing (eINP) and Software-Defined Networking (SDN). The concept of eINP differs from In-Network Processing (INP) because the latter is directly performed on NDs, whereas the former is performed on ENs connected via high-speed links to NDs. The rationale of eINP is to provide the network with additional capabilities for packet payload inspection and processing through edge computing, either on-premises or at MEC nodes. Although eINP is characterized by longer transmission and processing times than INP, it enables degrees of expressiveness that INP-enabled NDs cannot support at their current maturity stage. The overall infrastructure thus consists of a joint orchestration of eINP modules (which implement the computing capabilities, i.e., data aggregation, filtering, and encryption) across on- and off-premises ENs and SDN-enabled NDs for providing flow prioritization and steering.

In our previous work, we first designed an SDN-based middleware for dynamic rerouting and per-flow traffic prioritization in spontaneous wireless mesh networks [6]. Then, we explored data formatting of industrial traffic based on a standardized and widely accepted syntax for subsequent payload inspection and processing [7]. Subsequently, we proposed the concept of eINP along with the SDN-eINP interplay [8]. Lastly, we investigated the feasibility of eINP modules deployed through a container-orchestration system in industrial environments where strong mobility is present [9]. This article, which relevantly and originally extends our previous work, specifically focuses on MEC-integrated industrial environments, where on- and off-premises ENs are used in a coordinated way for enabling application-aware network traffic management. In our previous papers, network traffic was delimited within the industrial boundaries (on a locally owned and managed infrastructure); here, by concentrating on MEC integration in future Internet industrial scenarios, given that the considered resources are not only outside the industrial boundaries but also possibly owned by third parties, the confidentiality of information becomes critical. Indeed, in addition to symmetric cryptography, this work originally proposes to apply homomorphic cryptography in this context and describes the related tradeoffs in terms of performance and confidentiality.

The rest of the article is structured as follows. Section 2 introduces the related work. Section 3 discusses MEC-integrated industrial environments, focusing on the manufacturing sector. Section 4 discusses the drivers of the proposal. Section 5 outlines the main architectural components of the proposed solution. Section 6 provides implementation details, describes the testbed setup, and lays out performance results achieved on top of the implemented proof-of-concept prototype. Finally, Section 7 provides concluding remarks.

## 2. Related Work

This section lays out a literature review of the proposals sharing some commonalities with ours. Because the pillars of our work are SDN and eINP for enabling application-aware network traffic management in MEC-integrated industrial environments, we will discuss where edge computing, SDN, and INP (or a combination of them) are (jointly)

adopted in the literature. In this regard, we will first explore how industrial environments take advantage of edge computing. Then, we will investigate edge computing and SDN to support IIoT contexts. Subsequently, we will delve into the realm of INP-based proposals. Finally, we will conclude by pointing out the novelty of our work as compared to the reviewed literature.

Industrial environments adopt edge computing more and more frequently with the primary goal of bringing computation close to data sources, i.e., from the cloud to on-premises [10,11]. This may improve, among other things, data protection (by processing sensitive data on-premises rather than in the cloud) and real-time responsiveness (because latency at the edge is much lower than at the cloud). In this regard, ref. [12] proposes an architecture to support time-sensitive tasks and real-time data analysis at the edge whereas big data analytics tools reside in the cloud.

The European Telecommunications Standards Institute (ETSI) has defined MEC as a "system which provides an IT service environment and cloud-computing capabilities at the edge of an access network which contains one or more type of access technology, and in close proximity to its users" [13], provided a framework and reference architecture for it [14,15], detailed its technical requirements [16], and explored its integration with 3GPP 5G [17]. The role of MEC has also been explored in several industrial verticals [18]. For example, ref. [19] designs a multilevel MEC-compliant computing architecture that enables the unified management of fog/edge-computing resources in Industry 4.0. Instead, ref. [20] proposes a multitier MEC platform to meet the demanding requirements of IIoT applications and explores its pros and cons.

The adoption of edge computing and SDN [21,22] is gaining momentum. For instance, ref. [23] adopts SDN to handle the edge–cloud interplay in IIoT environments. Specifically, the authors relied on SDN to handle big data streams in an energy-aware and QoS-guaranteed fashion. Similarly, ref. [24] combines edge computing with SDN to support data streams with different latency constraints among IIoT devices. The SDN paradigm turned out to be effective even in wireless ad hoc scenarios, which are typically characterized by node mobility and heterogeneity [25]. For example, ref. [26] provides high-reliability and low-latency vehicle-to-everything communications by joining vehicular ad hoc networks and wireless cellular networks, wherein base stations are equipped with MEC technology. Instead, ref. [27] proposes an intelligent multiattribute routing scheme based on fuzzy logic, whereby buses and vehicles act as relay nodes.

Recently, INP, also known as in-network computing, has been proposed to support the execution on NDs of software modules that typically run on end hosts [28,29]. However, its adoption in industrial environments has not yet been widely investigated. A notable first proposal is [30], which proposes an SDN-based adaptive transmission protocol to support time-critical services by on-demand activating in-network functions for in-path caching and retransmission. An example of making edge computing and INP work together is found in [31], which offloads critical lightweight (sub)tasks to NDs while keeping the others on edge-computing resources.

Some INP-related proposals already provide forms of payload processing directly within NDs but with a limited degree of expressiveness. For instance, ref. [32] proposes PayloadPark (based on P4 [33]), which parks payloads in programmable switch memory, sends header-only packets to network-function servers (that primarily make decisions based on headers), and resembles them as such packets come back. Ref. [34] uses P4-enabled switches to aggregate packet sharing common headers into one and then to disaggregate the aggregated packet before reaching the destination. Ref. [35] programs P4-enabled switches to generate an alarm if MQTT messages convey values exceeding a given threshold.

Inspired by the work outlined above, we propose application-aware network traffic management in MEC-integrated industrial environments through the SDN-eINP interplay. Such an interplay makes the SDN side (how traffic flows traverse the industrial network, i.e., flow prioritization and steering) and the eINP one (how those flows are to be processed

along their path, i.e., data aggregation, filtering, and encryption) work synergically. The rationale of eINP is to provide the network with additional capabilities for packet payload inspection and processing through ENs connected to NDs via high-speed links to minimize communication overhead. If compared to related work, the adoption of the proposed eINP fuels NDs with ENs, thus making available general-purpose computing resources to offload payload inspection and processing. This, in turn, enables forms of payload processing at a higher degree of expressiveness that neither traditional nor INP-enabled NDs support at their current maturity stage. Such ENs may be deployed either on-premises, i.e., within the industrial boundaries, or off-premises, i.e., on MEC-enabled microdatacenters.

## 3. MEC-Integrated Industrial Environments

Nowadays, manufacturing companies are logically organized on three primary levels: shop floor, plant, and enterprise [36].

The shop floor level is mainly focused on industrial automation. As depicted in Figure 1, the primary components of the shop floor are industrial machines, Programmable Logic Controllers (PLCs), Human-Machine Interfaces (HMIs), IIoT devices, and AGVs. Industrial machines tend to have extremely long lifetimes (in the order of decades) and may implement different (proprietary) protocols. In addition, software upgrades may not always be possible, because manufacturers usually forbid software upgrades for safety reasons, or industrial machines may not support them at all. Instead, IIoT devices are characterized by a substantially shorter lifetime and usually communicate via well-known protocols. Then, PLCs are tailor-made devices for controlling manufacturing processes while coping with the harsh conditions affecting industrial environments, such as temperature surges, vibrations, and electrical noise. Human operators interact with and receive feedback from industrial machines through HMIs, specifically designed for improving operator decision making. In addition, the shop floor has been recently enriched by the deployment of AGVs, automating the process of moving products among different parts of the industrial environment.

The plant level regards the management of manufacturing processes. The critical component is the Manufacturing Execution System (MES). In particular, the MES receives instructions from operators about how industrial machines should behave, and then it transmits such instructions downward, i.e., toward the shop floor. As the requested manufacturing process goes into production, the MES matches the instructions provided by the operators against the actual actions taken by industrial machines. If the current actions do not match the desired actions, the MES applies corrective measures.

The enterprise level is about making decisions on how to run business operations. In this regard, decision makers rely on the Enterprise Resource Planning (ERP). The ERP collects information from the underlying business assets, such as supply chains, cash flows, customer orders, and production processes. Then, it processes such information by using business analytics to provide decision makers with an enterprise-wide picture of what is going on.

The Purdue model [37] is arguably the most common network implementation of such a logical structure. A pillar of the Purdue model is the concept of network segmentation. In particular, the Purdue model recommends a hierarchical approach that splits the industrial network into five layers. The lower three layers concern OT, whereas the upper two concern IT. The shop floor components crafting goods belong to layer 0/1. This layer relies on a time-sensitive network connecting industrial machines and PLCs. Then, layer 2 hosts devices that control the crafting processes (e.g., HMIs), whereas layer 3 includes those components that manage the manufacturing process as a whole (e.g., the MES). Finally, layers 4 and 5 consist of more IT-oriented functionality and facilities, such as Web servers, email servers, databases, and the ERP system, to name a few.

From a networking perspective, each layer of the Purdue model is supposed to be supported with differentiated performance levels. As a rule of thumb, the lower, the better. For example, because layer 0/1 must meet safety-critical requirements, the network is expected to provide high reliability and low latency (e.g., within 10 ms). However, layer

0/1 does not typically provide computing resources as abundantly as the upper layers. In this regard, layer 2 tends to provide worse network performance (e.g., within 25 ms) than layer 0/1, but it also potentially provides more computing resources and less stringent security requirements.

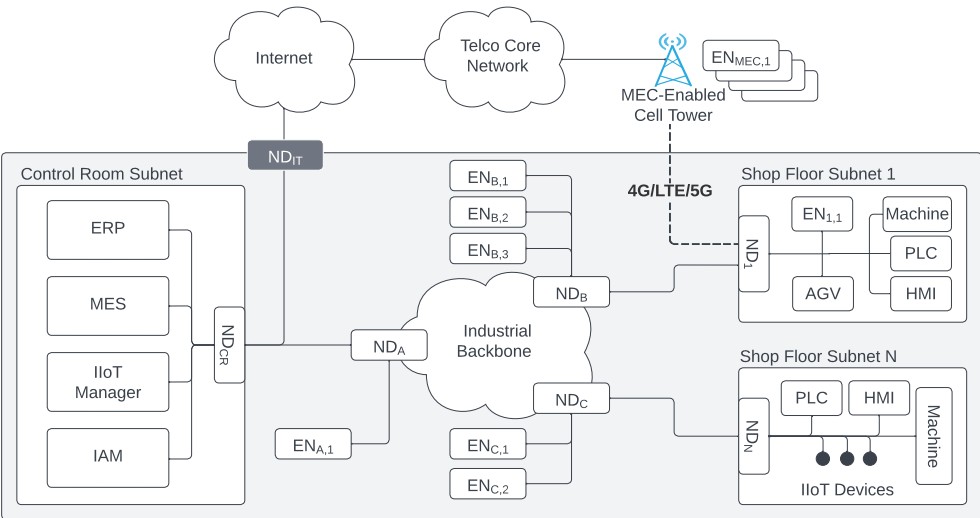

**Figure 1.** A MEC-integrated industrial environment. The industrial boundaries, depicted in light gray, include control room and shop floor subnets interconnected by the industrial backbone. MEC resources are outside the industrial boundaries, possibly available over cellular links at the shop floor level.

Furthermore, up-to-date industrial guidelines for cybersecurity (e.g., IEC 62443 [38]) recommend splitting the network topology into several shop floor subnets and a control room subnet. As Figure 1 shows, shop floor subnets consist of the shop floor components and an ND that works as a subnet gateway. The control room subnet comprises both plant and enterprise components, plus a subnet gateway. The subnet gateways are wired to the industrial backbone, but the shop floor ones may also be equipped with wireless interfaces, such as cellular. The network backbone connects such subnets through multiple communication channels to increase performance and fault tolerance. The outcome is a multihop, multipath topology providing several end-to-end connectivity opportunities with differentiated performance [39].

The definition of recent cellular standards enriched the industrial environments with advanced wireless communication and distributed computing capabilities [40–42]. In fact, the spread of 4G/LTE/5G technologies enabled wireless communications at increasing bandwidth with lower and lower latency. Moreover, the definition of the MEC standard provides the capability of deploying custom software components in the cellular operator network, close to industrial equipment. Let us stress that MEC nodes can be deactivated in an on-demand, cloud-like fashion.

Compared with traditional on-premises edge computing solutions deployed within industrial boundaries, MEC-based solutions suffer from higher latency. However, MEC-based solutions have the advantage that computing capabilities can be elastically managed if, when, and where required. Although traditional on-premises edge computing solutions achieve better latency than MEC ones, deploying a new EN requires purchasing, physically deployment, and configuration before it can host and run custom software modules.

Finally, let us note that on-premises edge and MEC-based solutions differ in trust levels. In the former, traffic management is completely performed within the industrial boundaries. Thus, there is (almost) no issue with confidentiality. In the latter, traffic flows traverse the cellular operator network (see the segment from the subnet gateway of shop floor subnet 1 to the MEC-enabled cell tower in Figure 1) and possibly the Internet. Therefore, data must be encrypted to ensure confidentiality. Two primary encryption options may

be considered. First, encryption may occur on-premises: this is the safest alternative because no cleartext data flow outside the industrial boundaries. Secondly, encryption may occur on the MEC node if the cellular operator can be considered a trusted party: this frees computing resources on-premises at the price of confidentiality over the cellular segment connecting the industrial environment with the MEC-enabled microdatacenter at the cell tower.

## 4. The Case for Application-Aware Network Traffic Management

The widespread deployment of IIoT devices has enriched industrial environments, delving into a quick shift toward smart factories. This has fostered an unprecedented volume of information, enabling disruptive use cases, such as mobile asset tracking, online remote reconfiguration, and predictive maintenance. In addition, the deployment of AGVs, in particular, and of relocatable industrial equipment, in general, has led to scenarios characterized by strong mobility. In such scenarios, dynamic and quick modifications in how traffic flows traverse the network, even at service provisioning time, cannot be considered extraordinary events. Therefore, compared with former industrial environments that traditionally crafted the same product(s) for long periods, there is now the need to manage frequent, abrupt, and sometimes even unpredictable movements of industrial equipment. Furthermore, modern industrial environments typically involve multiple industrial applications running concurrently and competing for shared resources. Such applications are fueled by the data produced on the shop floor level. The best way to deliver such data is therefore critical for guaranteeing the application requirements. This calls for application-aware network traffic management, whose objective is to manage traffic by considering application-level and application-specific requirements and characteristics.

The first step toward application-aware network traffic management is prioritizing mission-critical flows over non-mission-critical ones, thus programming the network infrastructure accordingly. In this regard, mission-critical flows must be delivered promptly and reliably, whereas non-mission-critical ones may be delayed or even partially dropped when necessary to free shared resources for the former. Flow prioritization concerns the capability of setting priority levels for flows based on the information they carry. Flow steering instead concerns the capability of programming the underlying network infrastructure, thus commanding the network behavior.

However, as huge traffic flows might traverse the network concurrently and abruptly migrate over time, more than flow prioritization and steering are needed for backing up application-aware network traffic management. There is also the need to be able to shape traffic flows, as they traverse the network by processing packet payloads. In fact, the raison d'être of data aggregation and filtering is to reduce the traffic passing through the network. A typical example is the aggregation of successive messages carrying temperature values in only one, which contains the average temperature of the aggregated messages. Another example is the filtering of messages carrying vibration values within the considered safety range, in order to transmit only out-of-range conditions. In both cases, the volume of data traversing the network decreases while not affecting the quality of information used by the supported applications. In contrast to flow prioritization and steering, which can be enforced by SDN-enabled NDs, data aggregation and filtering require general-purpose computing resources, going beyond traditional networking. Such resources are provided by ENs connected via high-speed links to NDs. This minimizes the overhead of transferring data back and forth between NDs and companion ENs while, to some extent, powering NDs with general-purpose hardware.

However, the on-premises resources might fall short in dealing with never-seen-before circumstances. In such cases, application-aware network traffic management might require peaks of resources beyond those available. To fill this gap, we consider MEC-integrated industrial environments, where subnet gateways at the shop floor level are equipped with cellular interfaces (e.g., 4G/LTE/5G) toward MEC-enabled cell towers. Not only MEC can nodes be de/activated elastically (meaning no up-front costs are required), but because they

are directly deployed on cell towers, telco operators can provide such resources with low latency (largely less than for any other virtualized resource at traditional geographically distant datacenters) to their users. Note that low latency is crucial when MEC nodes perform mission-critical tasks requiring a timely response.

However, steering traffic flows outside the industrial boundaries raises confidentiality concerns, which cannot be neglected in industrial environments. This calls for a fifth capability, i.e., data encryption. Because data encryption is CPU-intensive and must be enforced as flows traverse the network, it introduces important tradeoffs to consider at runtime. If the wireless communication link toward the MEC-enabled cell tower is trusted, only MEC resources could be used for encryption and the only data whose routes traverse the Internet would be required to be encrypted. On the other hand, if there is no security assumption over the wireless communication link toward the MEC-enabled cell tower, data encryption should occur on-premises. In this regard, note that (i) some legacy industrial machines might not implement any security by design, and software updates might be forbidden by manufacturers, (ii) some resource-constrained devices might not support any security protocol because of insufficient local resources, and (iii) some battery-powered devices might not implement security mechanisms for reducing energy consumption. In our proposal, we investigated symmetric and asymmetric algorithms, each with its pros and cons, to select the most suitable tradeoff depending on application requirements and the characteristics of the considered deployment scenario. Symmetric algorithms, for example, have the drawback of requiring the sharing of a secret key among the parties. Asymmetric algorithms, instead, overcome such drawbacks but are more computationally intensive and slower than symmetric ones. In particular, we explored homomorphic cryptography (a kind of asymmetric cryptography) because it allows the processing of encrypted data without first decrypting them. This, in turn, enables payload processing on off-premising edge computing resources without breaking confidentiality.In our proposal, several cryptography algorithms are available, e.g., a/symmetric and homomorphic, each with its own pros and cons, in order to select the most suitable tradeoff depending on application requirements and the characteristics of the considered deployment scenario. For example, symmetric algorithms are faster than asymmetric and homomorphic ones but have the drawbacks of a secret key shared by the parties. Homomorphic algorithms, instead, have the advantage of allowing computations on encrypted data without first decrypting them, but at the expense of higher latency.

In short, enabling application-aware network traffic management requires novel solutions to jointly orchestrate the networking and computing capabilities outlined above, with the most suitable dynamic tradeoffs tailored to the targeted deployment scenarios and supported industrial applications.

## 5. SDN-eINP Interplay for Application-Aware Network Traffic Management

This section lays out the architecture we designed to provide application-aware network traffic management through the SDN-eINP interplay. From a high-level perspective, the architecture consists of a control plane and a data plane (see Figure 2). In the following, we will break down those planes into fundamental components.

### 5.1. Control Plane

The control plane is responsible for making decisions about (i) how a flow traverses the industrial network (flow steering), (ii) its priority along the path (flow prioritization), and (iii) how that flow is to be processed (data filtering, aggregation, and encryption). The control room is where the control plane architectural components are located. Such components consist of an Industrial Application Manager (IAM), an SDN controller, and an eINP controller.

The IAM is logically built on top of the SDN and eINP controllers. Its primary task is to provide application-aware network traffic management by bridging the SDN and

eINP sides. To do so, the IAM takes advantage of the abstractions provided by each side, exploited as building blocks to enable the SDN-eINP interplay.

The SDN paradigm advocates for a clear-cut separation between who forwards packets and who makes routing decisions. The SDN controller is a logically (but not necessarily physically) centralized software entity that abstracts the underlying network infrastructure and provides the control logic to program the whole network. Specifically, it builds a network-wide view by gathering status statistics from NDs and dictates their behavior by pushing forwarding rules into them. Thus, NDs become dumb devices that make flow-based forwarding according to the instructions provided by the SDN controller. A flow is defined by rules matching a set of packet header values.

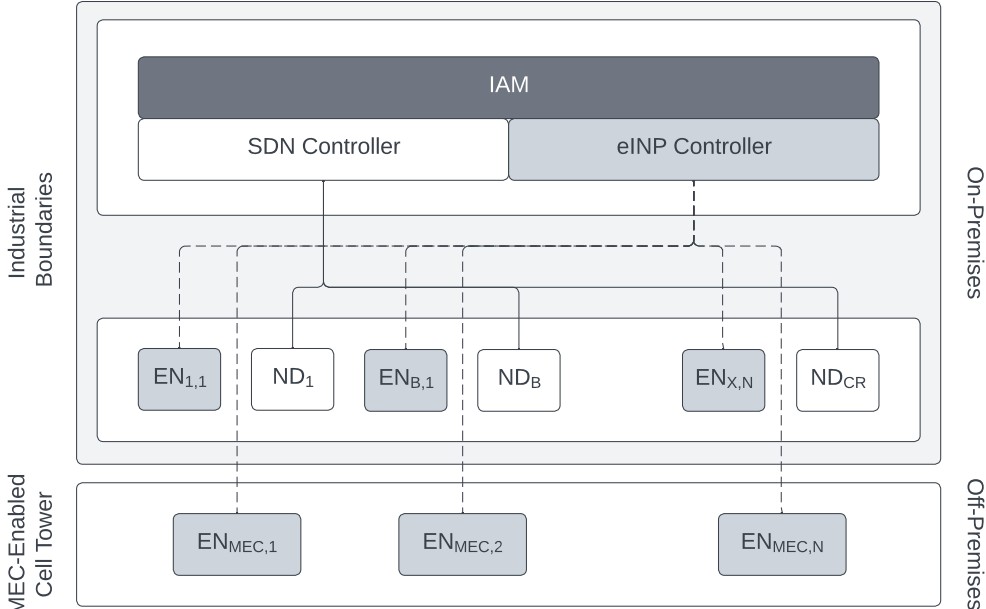

**Figure 2.** Architecture for enabling application-aware network traffic management. The architecture is based on the SDN-eINP interplay, where the IAM is the glue that makes the SDN and eINP sides work synergically. ENs are available both within (on-premises) and outside (off-premises) the industrial boundaries.

The eINP side abstracts computational resources made available by ENs and orchestrates software modules (packaged as containers) across them. The eINP controller clusters those resources in zones. The scope of a zone is to seamlessly orchestrate containerized eINP modules only across those ENs directly connected to a given ND. For instance, zone $B$ (see Figure 1) includes $EN_{B,1}$, $EN_{B,2}$, and $EN_{B,3}$, the only ENs directly connected to $ND_B$. The rationale behind that is to bring eINP modules as close as possible to the NDs where the target traffic flow has been scheduled to pass, avoiding messages traversing the industrial network back and forth to be processed.

The IAM relies on the network-wide view provided by the SDN controller to check the network status. This monitoring phase is an endless loop (see Figure 3), which gives the IAM the readiness to detect harmful communication patterns timely. An example of such a pattern is the following one. Let us consider a huge flow (e.g., an AGV sending vibration values to the MES) that has been scheduled along a given path. However, such a flow might undermine concurrent flows that (even partially) share the same path. In this case, the IAM may steer the flow toward an eINP module to be appropriately processed, as described in Figure 3. First, the IAM takes advantage of the abstractions provided by the eINP controller to retrieve the cluster-wide view. The cluster-wide view provides up-to-date information about the computing resources not reserved yet, the eINP modules currently running on the ENs, and how many resources such modules consume out of those reserved. Then, the decision-making phase takes place: IAM makes the SDN and eINP sides work synergically.

Based on the network- and cluster-wide views provided by the underlying controllers, the IAM jointly orchestrates "who runs where" and "who goes where." The former is about eINP module deployment, whereas the latter regards traffic flow steering. The eINP module deployment may not always be required, e.g., the target module is already running on an EN along the path. Moreover, it may affect multiple ENs, e.g., the eINP controller schedules multiple replicas of the target modules on different ENs. Then, the traffic flow steering takes place accordingly. Similarly, our traffic flow steering may affect multiple NDs in redirecting flows while considering their priority along the path.

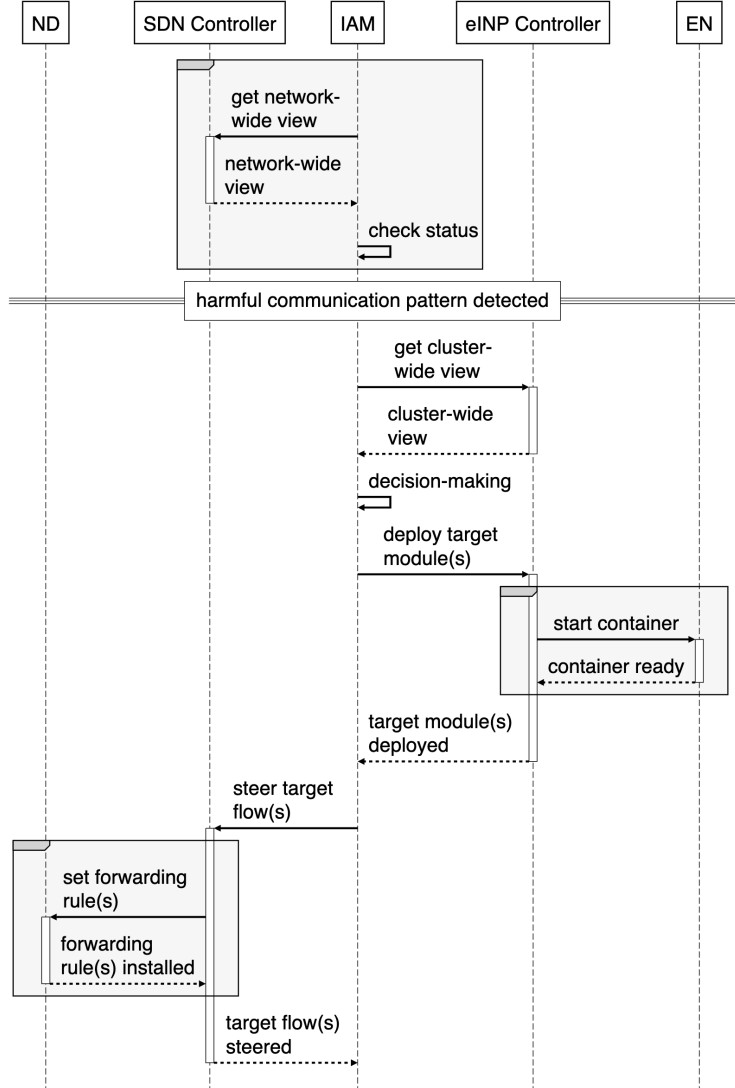

**Figure 3.** Sequence diagram detailing how the architectural components interact with each other to deal with a harmful communication pattern. The IAM jointly orchestrates the SDN and eINP controllers, which, in turn, dictate the behavior of NDs and ENs, respectively.

### 5.2. Data Plane

The data plane comprises backbone NDs as well as on- and off-premises ENs. It is worth remarking that NDs and ENs pursue different purposes, although working synergically. On the one hand, NDs enforces flow prioritization and steering. On the other hand, ENs applies processing functions to the data carried by messages belonging to such flows, such as data filtering, aggregation, and encryption. From an operational point of view, both NDs and ENs provide a kind of processing. The difference lies in what they process. Specifically, the former process packet headers, whereas the latter processes packet payloads.

As messages pass through an EN, the first step is payload deserialization. On top of that, an EN can perform a broad spectrum of actions at different degrees of expressiveness. Note that eINP is not limited to data aggregation, filtering, and encryption, which, to some extent, are our supported primitives. Other more sophisticated capabilities may be easily implemented on top of the primitives, such as pattern-matching detection and message duplication. For instance, an EN may detect hazardous temperature levels and duplicate those messages to alert the affected operators. Although message duplication may promptly improve information circulation, it also increases bandwidth usage. Capabilities that may help in this regard are data aggregation and filtering. Indeed, data filtering permits one to drop messages if the target field is evaluated as of little application interest (e.g., vibration values within a safe range), whereas data aggregation consists of, for example, averaging a target field carried in a certain number of consecutive messages that belong to the same flow and sending out a single message containing the average value. To do so, an EN queues incoming messages on a flow basis and performs the aggregation function as soon as the queue amounts to a given threshold. With data encryption, on the other hand, an EN may use a symmetric key to encrypt packet payloads of a target flow as they leave the sender, and then another EN may decrypt them before they reach the receiver.

*5.3. Running Example*

The starting point describes a stationary situation, highlighting two flows from different shop floor subnets (see Figure 4a). An AGV generates a flow (solid orange line) toward the MES, whereas an IIoT device produces another flow (dashed blue line) toward the IIoT manager. Although the running example considers only these two flows explicitly, other flows are expected to traverse the network concurrently. An eINP module performing data aggregation processes the AGV flow as close as possible to its source subnet. Because there are no ENs deployed within the source subnet, data aggregation occurs on $EN_{C,1}$, as the flow traverses the industrial backbone. At the same time, even the other flow goes through an eINP module. Specifically, the module performs data filtering, by dropping messages of more limited application interest. In this case, the shop floor subnet can exploit an EN; thus, data filtering is performed there.

Then, the AGV starts moving from one shop floor subnet to another, changing the wifi access point to which it is connected accordingly. As a result, its flow also needs a new path toward the MES. However, an eINP module for data aggregation is not available along the new shortest path (i.e., $ND_1$-$ND_B$-$ND_A$-$ND_{CR}$). On the one hand, the flow might go straight to the MES but with no aggregation in place. On the other hand, it might pass through the eINP module for data aggregation but via a longer path. In both cases, there is a waste of network resources. This triggers the IAM, in charge of checking for such harmful communication patterns. Let us assume that the IAM decides to (i) migrate the data filtering module on $EN_{B,1}$, (ii) steer the IIoT device flow accordingly, (iii) deploy a data encryption module (homomorphic cryptography) on $EN_1$, (iv) deploy a data aggregation module on $EN_{MEC,1}$, and (v) steer the AGV flow toward the MEC-enabled cell tower through the cellular interface of $ND_1$ (see Figure 4b).

In this particular example, the IIoT device flow traverses a resource-rich cluster zone along its path (i.e., $EN_{B,1}$-$EN_{B,2}$). This permits it to scale out and in module replicas dynamically. Because such a flow stresses the single replica available beyond a given threshold, the IAM starts a new module replica in the same cluster zone and thereby balances the incoming traffic accordingly (see Figure 4c).

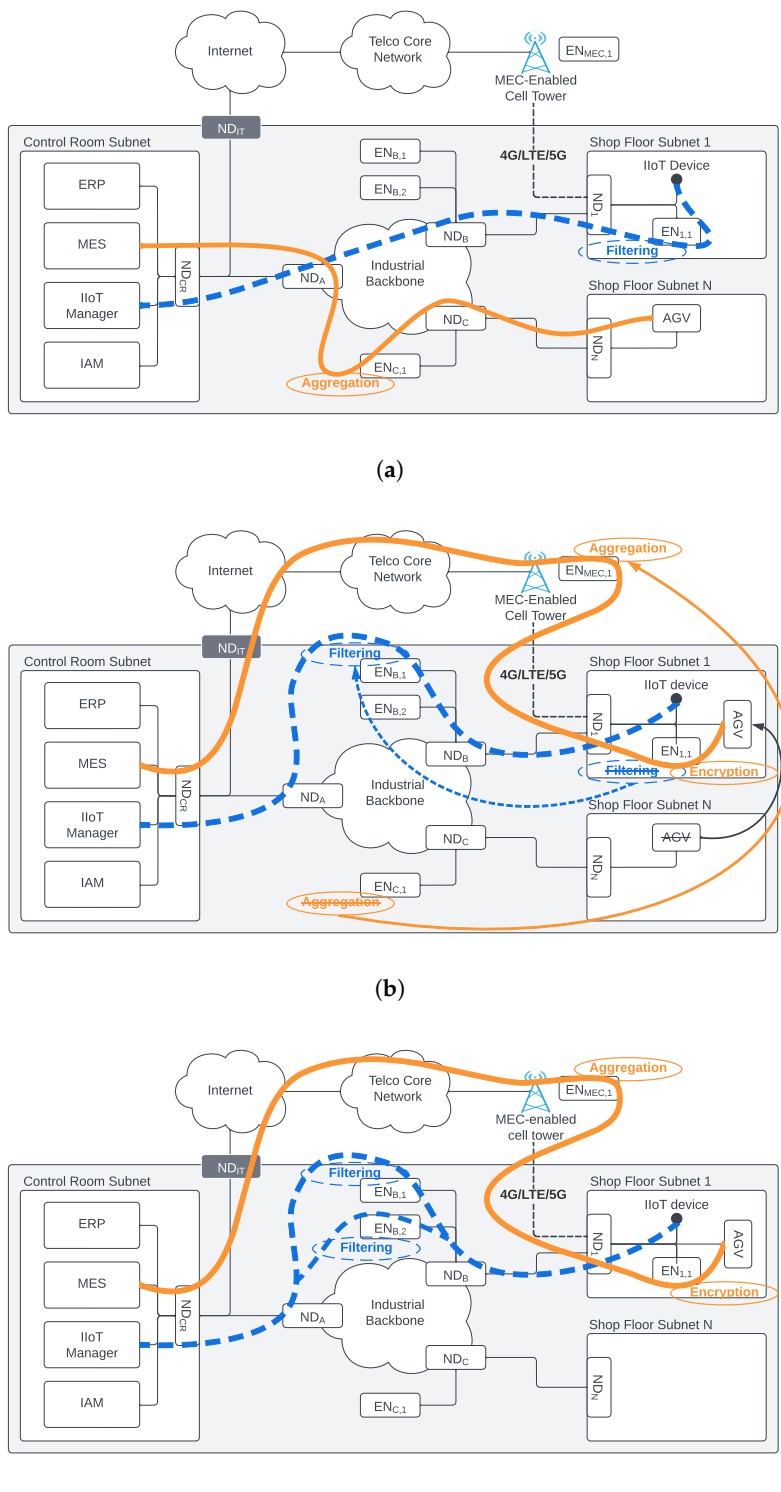

(**a**)

(**b**)

(**c**)

**Figure 4.** Running example of application-aware network traffic management in a MEC-integrated industrial environment. (**a**) Initially, the flows produced by the AGV (solid orange line) and IIoT device (dashed blue line) are processed on-premises. (**b**) Subsequently, the AGV changes to the shop floor subnet. This triggers the IAM, which decides to reconfigure the environment by also taking advantage of the MEC. (**c**) Because the flow produced by the IIoT device traverses a resource-rich cluster zone, additional replicas of the data filtering module are scaled out dynamically to better fit the demand.

## 6. Prototype Description and Evaluation

To assess the feasibility and efficiency of the proposed solution, we developed and experimentally tested a proof-of-concept prototype. In the following, we first describe the testbed setup in Section 6.1. Then, we detail experiments and analyze performance results in Section 6.2.

### 6.1. Testbed Setup

The testbed consisted of seven Virtual Machines (VMs) hosted on Amazon Web Services (AWS) Elastic Compute Cloud (EC2). Four of them made up a Kubernetes (v1.21.1) cluster involving a control plane and three workers. The workers acted as ENs while the control plane acted as the eINP controller. The ENs were split into two zones: "A" and "B." An additional VM mimicked an industrial machine, producing traffic flows at arbitrary rates. Another VM collected the messages generated by the industrial machine, acting as the MES. The last VM represented a technician's laptop. In this scenario, the technician was interested in getting warning messages only when specific circumstances occur, such as hazardous vibrations detected by the industrial machine.

We assumed that (i) zone "A" was the closest to the industrial machine, (ii) the SDN controller steered the traffic flows generated by the industrial machine through zone "A," and (iii) the eINP controller could not take advantage of resources available in zone "B" because it is not on the path. As the traffic grew, the control plane could decide to scale out horizontally (deploy more replicas of) the eINP module. Each cluster node ran Ubuntu 18.04 LTS and was equipped with two vCPUs and two GB of RAM. We chose CRI-O (v1.20.0) as the container runtime and Flannel (v0.14.0) as the Container Network Interface (CNI) plugin. The remaining VMs (industrial machine, MES, and technician's laptop) were based on Ubuntu 20.04 LTS and provided with one vCPU and one GB of RAM, respectively.

### 6.2. Experiments and Performance Results

The experiments varied along three dimensions. The first dimension was the rate at which the industrial machine sent out messages. In this regard, we set increasing rates from 2K to 10K Messages Per Second (MPS) in steps of 2K MPS. Each message was a UDP packet with a fixed payload size of 212 bytes.

The second dimension was the CPU usage to which the containerized eINP module was limited. Note that Kubernetes uses milli-CPU as the base unit for expressing amounts of CPU and allows specifying both CPU requests and limits [43]. Specifically, a CPU request states the minimum amount of CPU a container needs to run, and a CPU limit indicates the maximum amount of CPU a container can take while running. For all the experiments, we fixed the CPU request to 500 milli-CPU and tested the target eINP module with CPU limits of 500 milli-CPU and 1000 milli-CPU. When a container sets its CPU limit higher than its CPU request, it may benefit as follows. If a burst of messages causes a peak of computational demand, the container may successfully handle the incoming traffic by taking resources beyond its request. Otherwise, the container would run out of resources. Secondly, the horizontal autoscaler decides whether to scale replicas based on the ratio between the current CPU usage and the average of the given target values across running replicas, with a tolerance of 0.1 by default (please see [44] for more details). Given that a higher CPU limit allows a potentially higher CPU usage, the horizontal scaling is faster when peaks of computational demand occur. For all the experiments, the average target utilization was 80%, the minimum number of replicas was one, and the maximum number of replicas was four.

The third dimension was the eINP module under testing.

- Duplication & Aggregation (D&A): This module de/serialized incoming messages and detected mission-critical information with a probability of 5%. Such messages were duplicated and sent to the technician's laptop in a timely manner. Then, the module performed aggregation, reducing outgoing traffic by 90%.

- Symmetric Encryption (SE): This module de/serialized incoming messages, encrypted their contents with a Python implementation [45] of the AES algorithm [46], and forwarded them to the MES.
- Homomorphic Encryption (HE): This module de/serialized incoming messages, encrypted their contents with a Python implementation [47] of the Paillier algorithm [48], and forwarded them to the MEC node.
- Homomorphic Aggregation (HA): This module de/serialized incoming messages, queued messages on a flow basis, computed the average value of a target homomorphically encrypted field as soon as a queue grew up to ten elements, and sent out them to the MES.
- Homomorphic Decryption (HD): This module de/serialized incoming messages, decrypted their contents, and forwarded them to the MES.

To statistically support our findings, each module has been tested by repeating the experiments five times, and each run of an experiment captured a time window of 200 s (in Figure 5, each data point is the average value of five runs). Note that the following performance results are scenario-agnostic, thus providing the readers with insights about the eINP modules under testing, regardless of the employed industrial network topology.

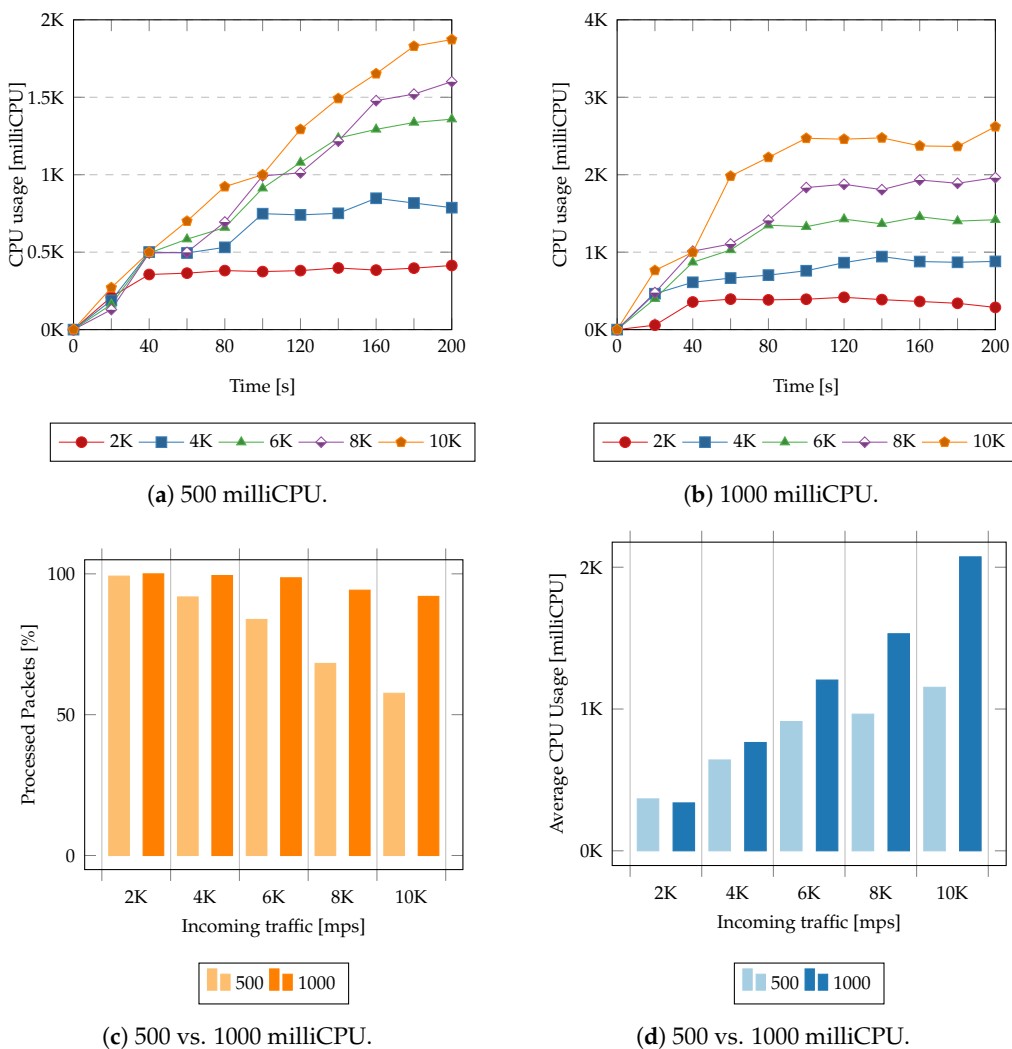

**Figure 5.** eINP module performing symmetric encryption with different milli-CPU limits.

While performing D&A, the average target utilization we configured (80%, meaning 400 milli-CPU) is not even triggered except for the worst-case scenario (10K MPS). It is worth mentioning that D&A is very efficient. In fact, aggregation dramatically decreases outgoing

messages, cutting down JSON serialization and forwarding costs. The key takeaway is that the eINP module correctly processed almost every packet at any incoming rate.

In contrast to D&A, SE is a CPU-intensive task. This makes horizontal autoscaling critical, even at relatively low message rates. As Figure 5a illustrates, the CPU usage grew to about 2000 milli-CPU at 10K MPS and above 1500 milli-CPU at 8K MPS. Given that Figure 5a depicts the experiments while CPU limited to 500 milli-CPU, it means Kubernetes scaled up to four replicas (i.e., the maximum allowed) to cope with such message rates. The comparison between Figure 5a,b points out the system responsiveness with different CPU limits. In Figure 5a, the lines about incoming rates of 6K, 8K, and 10K MPS still grow at the end of the time window (200 s), while all the lines reach a steady-state phase in Figure 5b. This is because replicas available at a given time could take extra resources (over the CPU request) while horizontal scaling occurred. As Figure 5c shows, replicas that can benefit from a higher CPU limit outperform those that cannot do it (500 milliCPU vs. 1000 milliCPU). For instance, the percentage of processed packets was 57.6% (500 milliCPU limit) against 92% (1000 milliCPU) at 10K MPS. Lastly, Figure 5d reflects this point in terms of average CPU usage.

As the reported results clearly show, horizontal autoscaling combined with a flexible resource cap was effective for performing SE, even while dealing with the worst-case scenario (i.e., 10K MPS). When dealing with HE, however, the performance was considerably different. In this regard, Figure 6a (note the logarithmic scale on the y-axis) shows the CPU usage while performing HE, HA, and HD on an incoming rate of 2K MPS. In this case, there were no resource requests or limits in place. This means eINP modules could take as many resources as they can from their host. Over the observed time window, the average CPU usage while performing HE (i.e., 890 milliCPU) was roughly 10 and 40 times higher than HA (i.e., 91 milliCPU) and HD (i.e., 23 milliCPU), respectively. In addition, Figure 6b shows the average per-message processing time, which is the time needed by the module to send a message after its reception. Specifically, the average per-message processing time was 10 times faster during HA (i.e., 35 ms) than HE (i.e., 349 ms). It is worth noting that while performing HA, only one out of 10 messages is homomorphically processed, whereas the processing time is negligible for the others. Lastly, the average per-message processing time while performing HD (i.e., 100 ms) was slower than HA but more than three times faster than HE.

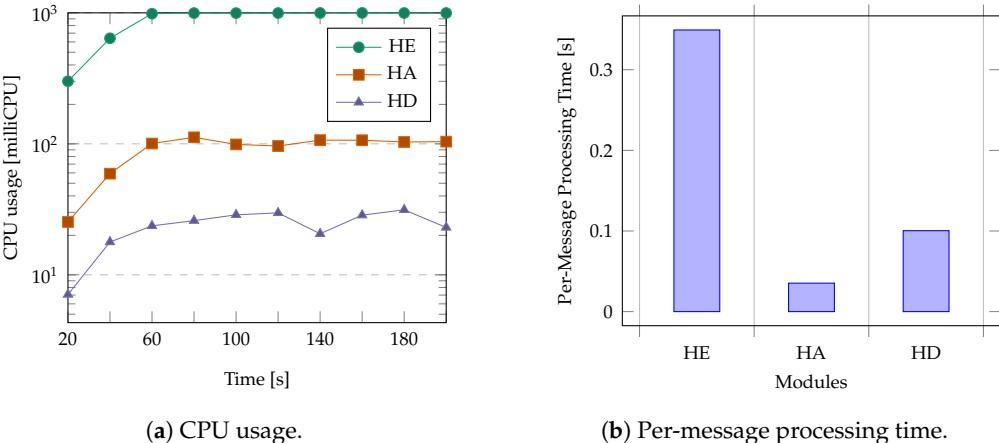

(**a**) CPU usage.          (**b**) Per-message processing time.

**Figure 6.** eINP modules performing homomorphic encryption, averaging, and decryption.

Based on the results we collected, we can state that in MEC-integrated industrial environments, eINP modules may be deployed both on- and off-premises. However, steering sensitive data on off-premises resources (i.e., MEC nodes) raises confidentiality issues. To fill that gap, we proposed providing the network with data encryption capabilities (i.e., SE and HE) through edge computing to enforce confidentiality even for those traffic flows originally sent out in cleartext. Although SE guarantees much better performance than



HE, SE requires that keys are shared between parties (i.e., industrial environment and MEC nodes) and that sensitive data must be decrypted before processing. This is not always a viable option, especially when confidentiality must be guaranteed all along the way. In this regard, HE could solve the issue because homomorphically encrypted data are left encrypted while processed, but it comes with a price. Our results suggest that HE is only practical if (i) a relatively high amount of edge computing resources are available on-premises, or (ii) a low MPS rate characterizes the target traffic flow (note the measured average per-message processing time of 349 ms while performing HE). Accordingly, the key takeaway is that there is no one-size-fits-all option, but the tradeoff between performance and confidentiality must be carefully evaluated depending on application-specific requirements and the characteristics of the targeted deployment environment.

In conclusion, data aggregation and filtering can dramatically reduce the network traffic traversing the industrial backbone while being computationally cheap. Ideally, such capabilities should reduce the traffic as much as possible without dropping relevant information for applications. The targets of data aggregation are those flows carrying data that are suitable for being the input of an aggregation function (e.g., the average), while the targets of data filtering are those flows carrying data that are not always relevant. Instead, data encryption targets those flows originally sent out in cleartext, thus ensuring confidentiality. In particular, homomorphic cryptography should be adopted only if (a) the target flow carries data suitable for being aggregated and/or filtered off-premises, (b) there is a strong confidentiality constraint requiring encrypted data all along the way, and (c) at least one of the two conditions above (i or ii) that make homomorphic cryptography practical is met.

## 7. Conclusions and Future Work

In this work, we explored application-aware network traffic management to deal with the unprecedented volume of data traversing industrial networks nowadays, while satisfying the requirements of network-intensive industrial applications. To this end, we designed an architecture based on the SDN-eINP interplay for enabling application-aware network traffic management in MEC-integrated industrial environments, where on-premises and off-premises edge computing resources are used in a coordinated way.

The rationale of eINP is to provide the network with additional capabilities (e.g., data aggregation, data filtering, data encryption, message duplication, and pattern-matching detection) for packet payload inspection and processing through edge computing. Because this implies additional delays, we developed a proof-of-concept prototype to quantify the performance of our solution while providing D&A, SE, HE, HA, and HD. We performed extensive in-the-field experimentation of the proposed solution: the reported performance results show the feasibility of our proposed approach and its tradeoffs in terms of performance and confidentiality. In particular, horizontal scaling revealed a promising mechanism for coping with CPU-intensive features, such as SE.

Such promising results foster the research activity towards more sophisticated orchestration strategies using predictive algorithms to activate eINP modules proactively based on movement patterns of mobile industrial components. We also plan to investigate how to make P4-enabled programmable switches and ENs work together.

**Author Contributions:** Conceptualization, M.F. and C.G.; methodology, M.F. and C.G.; software, M.F.; validation, M.F. and C.G.; formal analysis, M.F.; investigation, M.F.; resources, C.S.; data curation, M.F.; writing—original draft preparation, M.F. and C.G.; writing—review and editing, P.B., M.F., C.G. and C.S.; visualization, M.F. and C.G.; supervision, P.B. and C.S.; project administration, P.B. and C.S. All authors have read and agreed to the published version of the manuscript.

**Funding:** This work was partially supported by the European Union under the Italian National Recovery and Resilience Plan (NRRP) of NextGenerationEU, partnership on "Telecommunications of the Future" (PE0000001—program "RESTART").

**Data Availability Statement:** The data presented in this study are contained within the article.

**Conflicts of Interest:** The authors declare no conflict of interest.

**Abbreviations**

The following abbreviations are used in this manuscript:

| | |
|---|---|
| AGV | Automated Guided Vehicle |
| AWS | Amazon Web Services |
| CNI | Container Network Interface |
| D&A | Duplication & Aggregation |
| EC2 | Elastic Compute Cloud |
| eINP | Edge-powered In-Network Processing |
| EN | Edge Node |
| ERP | Enterprise Resource Planning |
| ETSI | European Telecommunications Standards Institute |
| HA | Homomorphic Aggregation |
| HD | Homomorphic Decryption |
| HE | Homomorphic Encryption |
| HMI | Human–Machine Interface |
| IAM | Industrial Application Manager |
| IIoT | Industrial Internet of Things |
| INP | In-Network Processing |
| MEC | Multi-access Edge Computing |
| MES | Manufacturing Execution System |
| MPS | Messages Per Second |
| ND | Network Device |
| PLC | Programmable Logic Controller |
| SE | Symmetric Encryption |
| SDN | Software-Defined Networking |
| VM | Virtual Machine |

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
