# Peer review of "Application-Aware Network Traffic Management in MEC-Integrated Industrial Environments"

_futureinternet, doi:10.3390/fi15020042_

Round 1

Reviewer 1 Report

The following comments to be addressed before consider this paper for publication:

1. the captions of the figures are brief and need to be elaborated.

2. The summary of the literature with highlighted limitations are presented. Which of these limitations addressed in the proposed work is to be mentioned.

3. This paper is theoretically weak, so theoretical analysis to be added to the paper.

4. there are several performance metrics in the literature, whereas the authors are requested to perform more metrics under different scenarios.

5. the reasons for enhancements of performance in the results section are explained. The limitations of the proposed work also listed.

6. For which applications this work is feasible? Provide the list of applications or use cases which are suitable for this work.

Author Response

Point R1.1: The captions of the figures are brief and need to be elaborated.

Response R1.1: We have included more exhaustive captions. Specifically, we revised the captions of Figures 1, 2, 3, 4, 4a, 4b, and 4c.

Point R1.2: The summary of the literature with highlighted limitations are presented. Which of these limitations addressed in the proposed work is to be mentioned.

Response R1.2: We have extended the last paragraph of Section 2 (Related Work) to point out which is the gap that our proposal has the ambition to fill. In particular, since eINP fuels NDs with ENs, it makes available general-purpose computing resources for payload inspection and processing. This, in turn, enables forms of payload processing at a higher degree of expressiveness (e.g., data aggregation, filtering, and encryption) that neither traditional nor INP-enabled NDs support at their current maturity stage. Then, in Section 6 we evaluated the cost for enabling such a higher degree of expressiveness, as well as we investigated the tradeoff between confidentiality and performance while using off-premises resources (i.e., MEC).

Point R1.3: This paper is theoretically weak, so theoretical analysis to be added to the paper.

Response R1.3: The aim of the paper is to present a novel architecture to show pros and cons of jointly providing networking (flow steering and prioritization) and computing (filtering, aggregation, and cyphering) capabilities for application-aware traffic management in industrial environments, where on-premises and/or MEC-based edge computing resources are available. We believe that the focus on architectural considerations allows readers to better understand when and how the identified modules can be fruitfully adopted to improve traffic flow QoS. A theoretical and in-depth analysis of the behavior of each single module would not be compatible with a minor revision and would significantly lengthen the manuscript; thus, we will take into account this valuable comment for separate future papers.

Point R1.4: There are several performance metrics in the literature, whereas the authors are requested to perform more metrics under different scenarios.

Response R1.4: We would like to stress that the measured and reported performance results aim at showing the behavior of our modules at different traffic/computation conditions despite the adopted scenario. In other words, the performance results in the paper are meant to guide readers regardless of the topology of the considered industrial backbone. We enhanced Section 6 (Prototype Description and Evaluation) to better clarify this aspect.

Point R1.5: The reasons for enhancements of performance in the results section are explained. The limitations of the proposed work also listed.

Response R1.5: We would like to thank the Reviewer for appreciating our discussion about the achievements and limitations of the currently reported performance results.

Point R1.6: For which applications this work is feasible? Provide the list of applications or use cases which are suitable for this work.

Response R1.6: We have added a paragraph at the end of Section 6 (Prototype Description and Evaluation) to sum up which kinds of target flows each computing capability targets, thus mentioning a typical application for each of them. Specifically, the targets of data aggregation are those flows carrying data that are suitable for being the input of an aggregation function (e.g., the average), while the targets of data filtering are those flows carrying data that are not always relevant. Instead, data encryption targets those flows originally sent out in cleartext, thus improving confidentiality.

Reviewer 2 Report

1.       In the abstract section, the author mentions the difference between eINP and INP and briefly introduces the basic principles of eINP. However, the innovation of this work compared to other work in the field did not be highlighted.

2.       Some relevant works in the same field are listed. However, the work is not summarized and categorized, which leaves the reader with no clear idea of the work in the field.

3.       In addition, existing works related to traffic prediction and management are not well discussed. Please refer to the following article and discuss it. [1] A Fuzzy Logic Based Intelligent Multi-Attribute Routing Scheme for Two-layered SDVNs, ” IEEE Transactions on Network and Service Management, 2022 

4.       Since compared with traditional solutions, MEC-based solutions have two disadvantages: 1) higher latency; 2) security and privacy issues. So why is MEC-based solution still being used for this work? Are there other reasons besides the elasticity to manage computing power.

5.       Please elaborate how ND performs flow prioritization and bootstrapping, and whether there are formulas/schematics to assist the reader's understanding.

6.       In the data encryption section, the authors mention two possible approaches: a/symmetric and homomorphic. Please specify the advantages of these two methods over other encryption methods and the circumstances under which they are applicable to this exercise.

7.       Please compare the similarities and differences of other methods to solve this problem and give corresponding references. Why did you choose these methods as comparison options?

Author Response

Point R2.1: In the abstract section, the author mentions the difference between eINP and INP and briefly introduces the basic principles of eINP. However, the innovation of this work compared to other work in the field did not be highlighted.

Response R2.1: We have extended the last paragraph of Section 2 (Related Work) to point out which is the gap we aim to fill. Since eINP fuels NDs with ENs, it makes available general-purpose computing resources for payload inspection and processing. This, in turn, enables forms of payload processing at a higher degree of expressiveness (e.g., data aggregation, filtering, and encryption) that neither traditional nor INP-enabled NDs support at their current maturity stage. The concept of application-aware network traffic management is based on the SDN-eINP interplay, where the former provides the networking capabilities whereas the latter provides the computing ones. This makes eINP a building block of application-aware network traffic management, whose rationale is described in Section 4 (The Case for Application-Aware Network Traffic Management).

Point R2.2: Some relevant works in the same field are listed. However, the work is not summarized and categorized, which leaves the reader with no clear idea of the work in the field.

Response R2.2: We have polished up Section 2 (Related Work). In the first part of the section, now, we explicitly mention which categories (i.e., edge computing, SDN, and INP) relate most to our proposal. Then, each paragraph is dedicated to those proposals that belong to a given category, or, in some cases, to those proposals that jointly belong to multiple categories. Lastly, the last paragraph of Section 2 points out how our proposal differs from the existing literature in the field. We strongly believe that this should facilitate the readability of the section and simplify the understanding of the existing related work in the field.

Point R2.3: In addition, existing works related to traffic prediction and management are not well discussed. Please refer to the following article and discuss it. [1] ” A Fuzzy Logic Based Intelligent Multi-Attribute Routing Scheme for Two-layered SDVNs, ” IEEE Transactions on Network and Service Management, 2022

Response R2.3: We have extended the paragraph about SDN in Section 2 (Related Work). The revised paragraph now discusses the paper titled “A Fuzzy Logic Based Intelligent Multi-Attribute Routing Scheme for Two-layered SDVNs.” We have also included the discussion of another work in the realm of SDVN: “A scalable and quick-response software defined vehicular network assisted by mobile edge computing,” which proposes SDN for vehicular ad hoc networks assisted by MEC nodes implemented at base stations.

Point R2.4: Since compared with traditional solutions, MEC-based solutions have two disadvantages: 1) higher latency; 2) security and privacy issues. So why is MEC-based solution still being used for this work? Are there other reasons besides the elasticity to manage computing power.

Response R2.4: We have added a comment in Section 1 (Introduction) about this particular point. The rationale of MEC in this context is that on-premises resources might not fit the demand in some cases (e.g., network traffic is temporarily too high for the employed industrial backbone). In such cases, a practical countermeasure is to steer some dynamically identified traffic flows outside the industrial boundaries over cellular links, such as 5G. The integration of 5G with MEC offers off-premises and elastically-provisioned edge computing in the proximity of industrial boundaries. Of course, the latency towards MEC nodes is anyway significantly lower than to the cloud, thus making the MEC a viable alternative even for demanding industrial scenarios. However, MEC brings its own challenges, such as higher latency than on-premises resources, and confidentiality issues (for the latter, see our solution with different forms of data encryption).

Point R2.5: Please elaborate how ND performs flow prioritization and bootstrapping, and whether there are formulas/schematics to assist the reader's understanding.

Response R2.5: We have further extended the paragraph about our previous work in Section 1 (Introduction) by mentioning  “SDN-Based Traffic Management Middleware for Spontaneous WMNs”. This work proposes an SDN-based middleware for dynamic rerouting and pre-flow traffic prioritization in spontaneous wireless mesh networks. Note that this work is a possible implementation of flow prioritization. Note that a primary feature of application-aware traffic management is that it is agnostic about how the specific networking and computing capabilities are actually implemented.

Point R2.6: In the data encryption section, the authors mention two possible approaches: a/symmetric and homomorphic. Please specify the advantages of these two methods over other encryption methods and the circumstances under which they are applicable to this exercise.

Response R2.6: In our paper we mentioned symmetric and asymmetric algorithms. Homomorphic cryptography falls within the realm of asymmetric algorithms. About symmetric cryptography, we focused on AES because in common practice it has replaced other symmetric algorithms such as DES and 3DES. We decided that usually adopted asymmetric algorithms, such as RSA, were not worth exploring because they are typically more computational intensive and much slower than symmetric ones. Thus, they can be extremely useful for exchanging symmetric keys, but are not typically used for ciphering relevant amounts of data. In the realm of asymmetric algorithms, we focused instead on homomorphic cryptography, since it allows simple operations on data with no need of first decrypting them. The rationale of homomorphic cryptography in the context of our work is to enable payload processing on MEC nodes without breaking confidentiality. We have modified Section 4 (The Case for Application-Aware Network Traffic Management) to better point out and clarify these aspects.

Point R2.7: Please compare the similarities and differences of other methods to solve this problem and give corresponding references. Why did you choose these methods as comparison options?

Response R2.7: Throughout the revision process, we have further elaborated on Section 2 (Related Work). As mentioned, the first part of the section now explicitly states which categories of proposals most relate to ours, while the last lines of Section 2 discuss how our proposal differs from the related state-of-the-art. What we propose, i.e., application-aware network traffic management, is based on multiple paradigms, i.e., edge computing, INP, and SDN. To the best of our knowledge, there are no other proposals in the current literature that use those paradigms in a joint fashion as we do. Therefore, we sought to lay out other proposals that rely on a single paradigm or on a simpler combination of them in a context close to the one that we target in the submitted paper.
